# Early Postoperative Pain Trajectories after Posterolateral and Axillary Approaches to Thoracic Surgery: A Prospective Monocentric Observational Study

**DOI:** 10.3390/jcm11175152

**Published:** 2022-08-31

**Authors:** Pascaline Dorges, Mireille Michel-Cherqui, Julien Fessler, Barbara Székély, Edouard Sage, Matthieu Glorion, Titouan Kennel, Marc Fischler, Valeria Martinez, Alexandre Vallée, Morgan Le Guen

**Affiliations:** 1Department of Anesthesiology and Pain Management, Hôpital Foch, 92150 Suresnes, France; 2Université Versailles-Saint-Quentin-en-Yvelines, 78000 Versailles, France; 3Department of Thoracic Surgery and Lung Transplantation, Hôpital Foch, 92150 Suresnes, France; 4Department of Research and Innovation, Hôpital Foch, 92150 Suresnes, France; 5Department of Anesthesiology and Pain Unit, Hôpital Raymond Poincaré, Assistance Publique Hôpitaux de Paris, 92380 Garches, France

**Keywords:** lung surgery, pain, trajectory, thoracotomy

## Abstract

Less-invasive thoracotomies may reduce early postoperative pain. The aims of this study were to identify pain trajectories from postoperative days 0–5 after posterolateral and axillary thoracotomies and to identify potential factors related to the worst trajectory. Patients undergoing a posterolateral (92 patients) or axillary (89 patients) thoracotomy between July 2014 and November 2015 were analyzed in this prospective monocentric cohort study. The best-fitting model resulted in four pain trajectory groups: trajectory 1, the “worst”, with 29.8% of the patients with permanent significant pain; trajectory 2 with patients with low pain (32.6%); trajectory 3 with patients with a steep decrease in pain (22.7%); and trajectory 4 with patients with a steep increase (14.9%). According to a multinomial logistic model multivariable analysis, some predictive factors allow for differentiation between trajectory groups 1 and 2. Risk factors for permanent pain are the existence of preoperative pain (OR = 6.94, CI 95% (1.54–31.27)) and scar length (OR = 1.20 (1.05–1.38)). In contrast, ASA class III is a protective factor in group 1 (OR = 0.02 (0.001–0.52)). In conclusion, early postoperative pain can be characterized by four trajectories and preoperative pain is a major factor for the worst trajectory of early postoperative pain.

## 1. Introduction

Effective postoperative analgesia plays a major role in the prevention of major morbidity and mortality [1] and is particularly important after thoracotomy since postoperative pain can increase the risk of respiratory complications, especially by limiting physical activity and physiotherapy [2]. Numerous advances have been made both in the surgical and anesthetic fields to reduce postoperative complications. Surgical techniques have been oriented towards less-damaging incisions (muscle-sparing posterolateral thoracotomy, axillary, or anterior approaches) or mini-invasive procedures (video-assisted thoracic surgery (VATS), and robot-assisted thoracic surgery (RATS) [3,4]. Postoperative analgesia techniques have also progressed with new techniques in loco-regional analgesia, which are substitutes for epidural analgesia (paravertebral nerve block, erector spinae plane block of the spine, cryoanalgesia, etc.) [5,6].

A better understanding of postoperative pain could make it possible to treat or even prevent it and, therefore, improve the postoperative experience [7]. Usually described as individual measurements of pain scores reported for each postoperative day, postoperative pain can also be described using the trajectory method, an original approach proposed by Chapman et al. [8] The pain trajectory is a longitudinal characterization of acute pain as a growth curve, normally resolving in intensity over days. The psychometric goal of growth curve modeling is to estimate the true dynamic course of acute pain resolution in each individual. With this simple linear model, each patient’s trajectory has two key features: the intercept or initial pain level, and the slope or the rate of pain resolution [8]. This makes it possible to classify a patient into a specific cluster (resolving pain over time, maintaining a constant level of pain over time, or increasing in pain over time) and to look for predictive factors.

This method was used to identify patient subgroups based on pain in various surgeries and notably after breast cancer surgery [9]. In a recent study, Vasilopoulos et al. identified five distinct postoperative pain trajectories from the data from day 1 to day 7 and characterized each group by age, gender, preoperative anxiety, and preoperative opioid consumption, but the patients enrolled had different surgeries [10].

Having conducted a prospective study regarding the risk of developing chronic pain after the axillary approach, a standard mini-invasive procedure, and posterolateral thoracotomies in thoracic surgery, we used the early postoperative data to establish the trajectories in post-operative pain and to identify their pre- and intraoperative predictive factors.

## 2. Materials and Methods

### 2.1. Ethics Approval

The present study reports the postoperative findings of a prospective, observational, single-center study performed in a tertiary care university hospital and was approved by the Ethical Committee Ile-de-France XI (n° 2013-A01641-44; Chairperson M. Catz) on 16 January 2014. The study was designed in accordance with the STROBE Standard for Observational studies) and published on the Clinical.trials.gov (accessed on 12 September 2014) website (NCT02237963).

### 2.2. Study Subjects

Patients were approached for consent before surgery and gave their written informed consent to participate. To be eligible, patients had to be at least 18 years old; French−speaking; and scheduled for lung surgery with a muscle-sparing posterolateral (PL group) or axillary thoracotomy (AX group) for lobectomy, pneumonectomy, wedge resection, bulla resection, or pneumothorax surgery for cancer or a non-cancer condition. Patients were excluded (a) if the planned procedure was a video-assisted thoracoscopic procedure, a partial or total pulmonary decortication, or a procedure extending to the chest wall; (b) if they had limitations of self-expression or communication by phone, or a severe psychiatric illness; and (c) if they had chronic thoracic pain before surgery (i.e., permanent pain over at least the three preoperative months). Pregnant and lactating women, and other vulnerable persons as defined by French regulation were not eligible.

### 2.3. Procedures

#### 2.3.1. Surgical Management

For each patient, PL thoracotomy or AX thoracotomy was chosen by the participating surgeons according to their experience.

For patients undergoing PL thoracotomy, the skin incision spanned the width of the latissimus dorsi. The latissimus muscle was completely divided, but the serratus anterior was spared if possible and reflected anteriorly. The chest was opened in the fifth intercostal space, and a rib retractor was systematically used.

The incision for the AX approach extended approximately 7 cm caudal from just below the axillary hairline along the anterior border of the latissimus. The latissimus was completely spared and did not require mobilization. The insertions of the serratus anterior muscle on ribs 4 and 5 were dissected from these ribs, allowing the muscle to be lifted from the chest wall to allow the intercostal incision to be made in the fourth intercostal space. A rib retractor was only used if required by the surgeon.

For both approaches, the intercostal incision was extended far anteriorly and posteriorly beyond the limits of the skin incision.

A small portion of the posterior sixth rib was occasionally resected (shingled) during PL thoracotomy. Both types of incisions were closed with three or four interrupted pericostal or intracostal sutures. A running absorbable suture was used on the muscle, subcutaneous, and skin layers of the PL thoracic incision and on the subcutaneous and skin layers of the axillary thoracic incision.

Postoperative chest tube management was at the discretion of the surgical team.

#### 2.3.2. Anesthetic and Pain Management

Anesthesia and pain management were standardized for all patients.

Anesthesia was total intravenous anesthesia by propofol and remifentanil, the doses of which were titrated according to hemodynamic stability and to maintain a Bispectral Index between 40 and 60, a range corresponding to the desired depth of anesthesia.

The usual pain management was thoracic patient-controlled epidural analgesia associated with co-analgesics (acetaminophen, nefopam, and nonsteroidal anti-inflammatory drugs in the absence of contraindication) given IV first and then orally as soon as possible. The epidural catheter was inserted before surgery at the T5–T6 or T6–T7 level; a first bolus of ropivacaine 0.375% mixed with 5 µg/mL (3 to 5 mL) of sufentanil was administered prior to anesthetic induction and was followed by a 5 mL/h infusion of 0.2% ropivacaine and 0.5 µg/mL of sufentanil. As soon as the patient was able to use the patient-controlled function, 3 mL boluses with a refractory period of 20 min were provided. In the case of contraindication to epidural analgesia (e.g., coagulopathy, systemic infection, or spinal disease), a paravertebral catheter was placed during surgery and allowed for 0.375% (3 to 5 mL/h) ropivacaine infusion. In the case of failure of epidural analgesia despite routine procedures, intravenous morphine patient-controlled analgesia or oral morphine was started. As specified in our institutional protocol, the epidural catheter was removed after the last chest tube was removed, typically on postoperative days 3 to 5, and oral analgesics (opioids and acetaminophen) were used alone.

Post-operative analgesia was given by nurses who specialize in pain management. This allowed for therapeutic adaptation.

### 2.4. Data Collection and Outcomes

Data were collected at inclusion, during surgery, and during the postoperative period. A complete assessment was performed on the 6th ± 1 postoperative day.

Demographic variables, history of cancer and tobacco use, pain, and anxiety were collected at inclusion. Regarding pain, patients were asked to respond to the question “Do you regularly suffer from pain?”. In the case of a positive response, they provided a self-assessment of pain at rest during the visit and the mean value of pain score during the prior week using an 11-point numerical rating scale (NRS) from 0 = no pain to 10 = maximum imaginable pain. They were also asked to locate the pain and to report their painkiller use using the World Health Organization analgesic classification. A neuropathic pain was defined by a DN4 score ≥ 4 [11]. Patients’ self-reporting of anxiety also used NRS with 0 = no anxiety and 3 = maximal imaginable anxiety.

The duration of anesthesia and surgery as well as the particularities of the surgical techniques were noted.

Postoperative pain was assessed each day from day 0 (day of surgery) to day 6 ± 1 at rest, when coughing and during ipsilateral shoulder mobilization using NRS [12]. The latter was the main outcome that was used to establish the trajectories of post-operative pain after thoracic surgery regardless of the surgical approach (PL thoracotomy or AX thoracotomy).

In addition, at day 6 ± 1, the postoperative analgesia technique; the DN4 score; the skin anesthesia or hypoesthesia around the scars using a 10 g Von Frey filament on three vertical measures at the midclavicular, midaxillary, and scapular lines; the total length of the scar; and patient satisfaction with pain management using a NRS from 0 = totally disappointed to 10 = totally satisfied were recorded.

Finally, the duration of postoperative stay and postoperative complications were collected using the Clavien classification [13].

### 2.5. Statistical Analysis

#### 2.5.1. Number of Patients to Be Included

The number of patients to be included was calculated using the occurrence of chronic pain after PL and AX thoracotomies as the main outcome. When planning the study as a randomized controlled trial, the sample size was calculated based on a preliminary report from our group showing a 48% prevalence of chronic pain one year after a posterolateral thoracotomy [14]. It was hypothesized that post-thoracotomy pain syndrome prevalence would be 40% lower after an AX thoracotomy. However, since there were about twice as many surgeons performing posterolateral thoracotomy than surgeons performing AX thoracotomies, it was anticipated that the ratio of patients in the two arms would be about 2 to 1. Based on a 2-sided Fisher’s exact test, with an alpha risk of 0.05, group sample sizes of 103 in the smaller group and 206 in the larger one achieve 90% power to detect a 0.60 smaller to larger ratio (i.e., 0.29 for the AX group and 0.48 for the posterolateral group) for the primary outcome. In April 2016, following a modification of our surgical practice, it appears that there were as many AX procedures as PL procedures. We therefore decided to stop our inclusions at 196 patients.

#### 2.5.2. Statistical Methods

All statistical analyses were conducted on an intention-to-treat basis.

Categorical variables are presented as number (proportion); continuous variables are presented as median (25th percentile–75th percentile).

The statistical analysis was carried out in three steps: comparison of patients according to their surgical incision, identification of postoperative pain trajectories, and comparison of patients according to their pain trajectories.

Comparisons of variables between patients who had a PL or an AX thoracotomy used a Chi2 or a Fisher test for categorical variables and a Mann–Whitney test for continuous variables. Mixed model repeated measures were used to compare pain scores at mobilization during the early (day 0 to day 5) postoperative days. The surgery group and the days were used as fixed factors.

In the primary analysis, to identify clusters or subgroups of patients with similar progressions, defining trajectories of pain after surgery (measured from day 1 to day 5), a mixed ascending hierarchical classification was implemented with PROC CLUSTER in SAS software (SAS Institute). Each patient was clustered into the trajectory group to which they had the highest posterior probability of membership. First, a principal component analysis (PCA) was applied on postoperative pain variables. Then, factor axes were used to determine a dendrogram using a mixed ascending hierarchical classification. The dendrogram was cut using the Euclidean distance between each point with the Ward method to identify the best number of subgroups (i.e., trajectories). Comparisons of variables between all trajectories used a Chi2 or a Fisher test for categorical variables and a Kruskal–Wallis test followed by a Dunn’s multiple comparisons test. Finally, a multivariable multinomial logistic model was performed to identify risk factors for trajectories. All factors associated with clusters in the univariate analysis with *p*-values < 0.20 were included in the multivariable model.

All statistics tests were two tailed, and P was considered significant when the *p*-values < 0.05.

Analyses were performed using SAS 9.4 (SAS Institute Inc., Cary, NC, USA) and R software (version 3.1, R Foundation for Statistical Computing, Vienna, Austria) using the application GMRC Shiny Stat (Strasbourg, France, 2017).

## 3. Results

### 3.1. Participants

Two hundred and fifteen patients were approached from July 2014 to November 2015, with one hundred and ninety-six finally included. Ninety-six patients underwent surgery through an AX incision, and one hundred underwent surgery through a PL incision. Initially, 25 patients (13 in the AX group and 12 in the PL group) were excluded from analysis due to missing pain scores. However, after imputation of the missing data, ten patients (six in the AX group and four in the PL group) were retrieved for data analysis. The analysis concerned 89 patients in the AX group and 92 patients in the PL group (Figure 1).

### 3.2. Comparison of Variables between the PL Group and the AX Group

There was only one difference between the patient characteristics at inclusion in the AX and in the PL groups; patients in the AX group were younger, with a median age of 62 years old, whereas patients in the PL group had a median age of 66 (*p* = 0.041) (Table 1).

During the intraoperative period, differences between the two groups were related to surgical particularities (surgical retractors and rib fracture more common in the PL group (*p* < 0.001 and *p* = 0.004, respectively), and latissimus dorsi preservation and transcostal suture more frequent in the AX group (*p* < 0.001)) (Table 2).

Pain scores for mobilization of the ipsilateral shoulder from the day of surgery to postoperative day 5 were similar in both groups (*p* = 0.83, Figure 2).

Most patients received thoracic epidural analgesia. Pain scores for cough and pain scores for mobilization of the ipsilateral shoulder were significantly lower in the AX group (respectively, medians of 3 and 2 in the AX group vs. medians of 4 and 3 in the PL group, with *p* = 0.019 and *p* = 0.035, respectively). Height of the hypoesthesia area around the scar was lower in the AX group (*p* = 0.016). Finally, the scar length was significantly greater in the PL group: median size 17 cm vs. 8 cm in the AX group (*p* < 0.001) (Table 3).

### 3.3. Pain Trajectories

The best-fitting model included four pain trajectory groups, with a combination of linear and quadratic trajectory groups. Those who partly or fully resolved their pain over five days had negative slopes, and those who demonstrated a pattern of increasing pain over days had positive slopes. Pain trajectory group 1 corresponds to patients with permanent significant pain from day 0 to day 5 (29.8% of the patients), pain trajectory group 2 corresponds to patients with moderate pain from day 0 to day 5 (32.6%), pain trajectory group 3 corresponds to patients with steep decreases in pain across the five days after surgery (22.7%), and pain trajectory group 4 corresponding to patients with steep increases in pain across the 5 days after surgery (14.9%) (Figure 3).

### 3.4. Comparison of Variables between the Trajectory Groups

Some patient characteristics at inclusion differed between trajectory groups, especially between trajectory group 1 and trajectory group 2: younger patients (*p* = 0.031), more patients with neuropathic pain (*p* = 0.013), and more patients having taken at least one analgesic medication before surgery (*p* = 0.022) in group 1. The percentage of preoperative painful patients was greater in group 1 than in the other groups (Table 4).

Intraoperative variables were similar regardless of trajectory group (Table 5).

Of interest, scar lengths did not differ from one group to another (Table 6).

### 3.5. Predictive Factors of Pain Trajectories

Variables in the multinomial logistic regression were age, preoperative state of health qualified by the ASA score, presence of preoperative pain, use of at least one pain medication, and preoperative anxiety. Three entities relevant to thoracic surgery were also analyzed: rib fractures, transcostal suture, and scar length. This corresponds to the variables reaching the defined threshold for inclusion (*p* < 0.20). The preoperative neuropathic component of pain was not included in the analysis because of the large number of missing values. The permanent non-significant pain trajectory group (group 2) is the reference group for multinomial logistic regression.

Some predictive factors allow for differentiation between trajectory groups 1 and 2. Risk factors for permanent pain are the existence of preoperative pain (OR = 6.94, 95% CI (1.54–31.27)) and scar length (OR = 1.20 (95% CI (1.05–1.38))). In contrast, ASA class III is a protective factor in group 1 (OR = 0.02, 95% CI (0.001–0.52)) (Table 7).

No predictive factor could allow for differentiation between trajectory group 2 and groups 3 or 4.

## 4. Discussion

### 4.1. Main Results of Our Study

To our knowledge, this is the first description of pain trajectories during the early postoperative period after lung surgery. For patients operated by PL or AX approaches using a mini-invasive technique like video-assisted thoracoscopic surgery, in terms of early complications, pain, performance status, and quality of life [15], four pain trajectories can be identified after lung surgery: constantly high, constantly low, decreased, and increased pain trajectories over time. Moreover, we identified two risk factors (preoperative pain and scar length) and a protective factor (ASA class III) for the worst pain trajectory when compared to the best one.

### 4.2. Interpretation

Defining acute postoperative pain as a trajectory rather than as one or more simple point estimates of intensity increases the information yield of pain assessment and improves measurement precision [8,16]. Thus, it must be underlined that a high percentage of patients experience pain during the early postoperative period despite the analgesic techniques put in place and followed-up by nurses who specialize in pain management. This percentage is 44.7% when adding patients with permanent significant pain from day 0 to day 5 (group 1) and patients with steep increases in pain over these days (group 4). Few authors have reported early postoperative pain trajectories. Althaus et al. pooled data of patients having undergone orthopedic surgery, general surgery, visceral surgery, and neurosurgery and reported three types of pain trajectory: little initial pain on the first postoperative day with further pain resolution (57% of the patients), severe pain with a high rate of pain resolution (30%), and permanent high pain intensity (13%) [17]. Five trajectory groups were reported by Vasilopoulos et al. after pooling data of patients having undergone major orthopedic, urologic, colorectal, pancreatic/biliary, thoracic, or spine surgery. Four trajectories identified patients with low (7% of the patients), moderate-to-low (24%), moderate-to-high (46%), and high pain (17%) over time; one trajectory corresponded to patients with drastically decreasing postoperative pain (6%) [10]. It is difficult to compare the distribution of patients according to trajectories between this study and ours. Thus, as in our study, Vasilopoulos et al. found that approximately one in two patients can be categorized as having a painful experience despite the use of many preoperative nerve blocks. However, postoperative pain treatment is not detailed, as pointed out by Kehlet and Foss [18].

There are numerous studies of the risk factors of postoperative pain using pain as the main outcome. Ip et al., studying 48 eligible studies with 23,037 patients, identified preoperative pain, anxiety, age, and type of surgery, but not gender, as significant predictors for postoperative pain [19]. A similar study by Gerbershagen et al. reported that female gender, younger age, and preoperative chronic pain are associated with higher postoperative pain in a large multicenter cohort of patients having had heterogeneous surgical procedures [20]. More recently, a systematic review and meta-analysis identified nine predictors of poor postoperative pain control, with odds ratios ranging from 1.02 to 2.32: younger age, female gender, smoking, depression, anxiety, sleep difficulties, high body mass index, presence of preoperative pain, and use of preoperative analgesics [21].

Regarding the specific question of postoperative pain after lung surgery, the work carried out by Kwon et al. must be underlined since it compared pain after robotic, video-assisted thoracoscopic surgery and open anatomic pulmonary resection and reported that female gender, younger age, and baseline narcotic use were associated with acute postoperative pain [22].

Replacing the measurement of postoperative pain with the determination of pain trajectories has little effect on the determination of risk factors. Thus, Vasilopoulos et al. also reported that risk factors for the stable moderate-to-high and high pain groups are female gender and young age while preoperative high anxiety and depression and greater pain behaviors and pain catastrophizing were linked to the stable high group [10]. After lung surgery, we found dissimilar predictive factors, preoperative pain and scar length, for painful trajectories and a protective factor, ASA class III. We did not find an association between female gender and increased postoperative pain intensity, as previously shown across a variety of surgical procedures [20,23,24], but as we have seen above, other studies did not find this factor [19]. Similarly, we did not find that young age is linked to increased postoperative pain, as previously reported [10,19,20,22], although van Dijk et al. reported in a large and heterogenous international cohort that postoperative pain decreases with increasing age, but these authors qualified this relation as small and of questionable clinical significance [25].

Among behavioral factors that have been reported as strongly associated with acute postoperative pain intensity, preoperative anxiety is highlighted [10,19,21]. This point is unfortunately not confirmed in our study, perhaps because more than two thirds of the patients were anxious preoperatively. This underscores the importance of studying risk factors in a homogeneous patient population with respect to their pathology and surgical procedure.

### 4.3. Strength and Limitations of the Study

The strength of our study is that it was conducted under real-life conditions, with the data collected reflecting the vagaries of management, including the successes and failures of surgical procedures and analgesia techniques.

On the other hand, our study suffers from several important limitations.

As specified before, this study is an ancillary study from another that studied the development of chronic pain after PL and AX approaches for thoracic surgery.

This study established four clusters describing postoperative pain by mixing patients operated on by PL thoracotomy or by AX thoracotomy. Posterolateral thoracotomy is increasingly being replaced by video-assisted thoracoscopic surgery and robot-assisted thoracic surgery, which result in less acute pain [22]. On the other hand, AX thoracotomy is rarely performed at present but represents a model of mini-invasive surgery. Numerous patients received thoracic epidural postoperative analgesia, preventing an analysis of the role of this technique in the prevention of postoperative pain.

Our study did not include randomization for the choice of approach because the surgeons considered it unethical to use approaches they were unfamiliar with. It was decided that participating surgeons would use the approach they routinely used and were familiar with. Such conduct is usual since most studies comparing PL thoracotomy and VATS do not include randomization.

Our preoperative screening of potential factors linked to postoperative pain was obviously incomplete. The evaluation of anxiety was rather short, and we did not evaluate psychological factors such as depression, pain coping and catastrophism, and sleep disorders. A genetic study is also missing.

### 4.4. Generalizability

Our study combined patients having had a large surgical approach and a mini-invasive one. These two approaches are being progressively discarded and replaced by video-assisted thoracoscopic surgery and robot-assisted thoracic surgery; our results cannot be generalized to the whole lung surgical population.

Moreover, most of our patients benefited from a thoracic epidural postoperative analgesia. This was the choice of our team at the time of this study, but this strategy has evolved towards less-invasive techniques [5,6].

## 5. Conclusions

In conclusion, we identified four trajectories for early postoperative pain after lung surgery performed by posterolateral and axillary approaches. The major risk factor for permanent pain trajectory is preoperative pain, which should lead to a more aggressive pre-operative management unless it represents a constitutive factor, perhaps genetic. We confirmed the benefit of mini-invasive surgery since scar length is also reported as a risk factor for postoperative pain. On the other hand, it is difficult to explain why ASA class III is a strong protective factor unless it is related to more careful management.

## Figures and Tables

**Figure 1 jcm-11-05152-f001:**
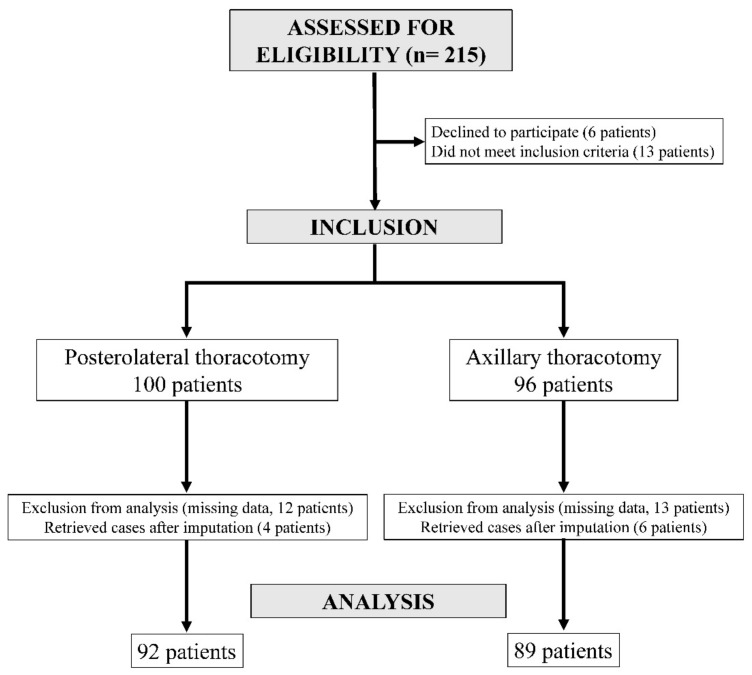
Flowchart.

**Figure 2 jcm-11-05152-f002:**
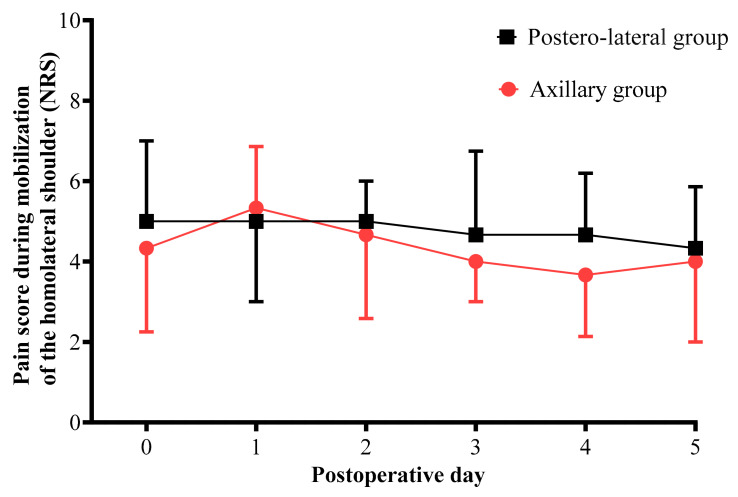
Comparison of pain score for mobilization of the ipsilateral shoulder measured from the day of surgery to postoperative day 5 for each surgical technique. Black boxes represent median NRS in the posterolateral thoracotomy group. Red circles represent median NRS in the axillary thoracotomy group. Pain was evaluated using an 11-point numerical rating scale (NRS) from 0 = no pain to 10 = maximum imaginable pain.

**Figure 3 jcm-11-05152-f003:**
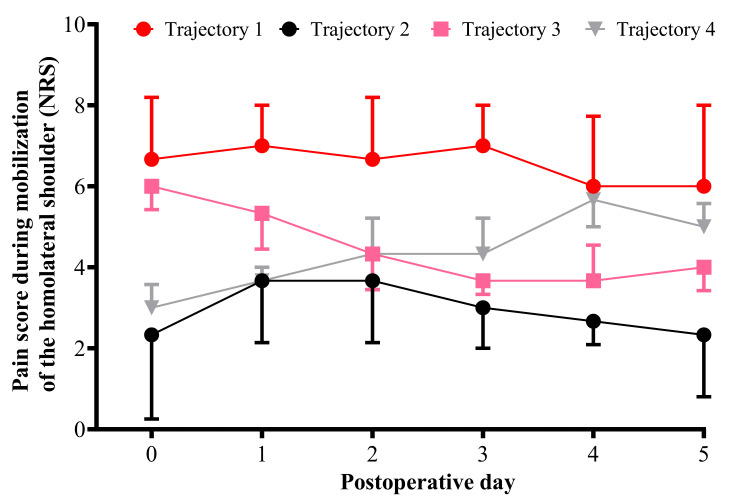
Comparison of pain score for mobilization of the ipsilateral shoulder measured from the day of surgery to postoperative day 5 for each trajectory. For each trajectory the median pain score is represented (numeric rating scale: 11-point scale for patient self-reporting of pain from 0 = no pain to 10 = maximal imaginable pain). The black line represents median pain scores in cluster 1 (permanent significant pain group). The red line represents median pain scores in cluster 2 (permanent moderate pain group). The pink line represents median pain scores in cluster 3 (decreasing pain group). The gray line represents median pain scores in cluster 4 (increasing pain group).

**Table 1 jcm-11-05152-t001:** Patient characteristics at inclusion in posterolateral and axillary groups.

	Posterolateral Thoracotomy n = 92	Axillary Thoracotomy n = 89	*p* Value
Age, years	66 (57–72)	62 (53–69)	0.041
Sex, Male/Female	50 (54.3)/42 (45.7)	42 (47.2)/47 (52.8)	0.374
Body mass index, kg/m^2^	25 (22–28)	24 (22–27)	0.227
ASA			0.129
I	5 (5.4)	13 (14.6%)	
II	68 (73.9)	59 (66.3%)	
III	19 (20.6)	17 (19.1%)	
Cancer	74 (80.4)	71 (79.8)	1
History of tobacco use	63 (75.0) {84}	60 (76.9) {78}	0.855
Current anxiety *	64 (69.6)	67 (81.7) {82}	0.079
Painful patients	42 (45.6)	33 (37.1)	0.291
Pain on inclusion day **	1 (0–3) {41}	2 (0–3) {29}	0.997
Mean pain during the prior week **	3 (2–6) {39}	4 (2–5) {28}	0.639
Thoracic location of pain	10 (25.0) {40}	4 (12.9) {31}	0.242
Neuropathic pain ***	8 (12.5) {64}	5 (9.1) {55}	0.769
Analgesic used by all patients ****	{66}	{59}	
At least one	27 (40.9)	17 (28.8%)	0.191
Step 1	24 (36.4)	17 (28.8%)	0.446
Step 2	9 (13.6)	2 (3.4%)	0.058
Step 3	2 (3.0)	0 (0.0%)	0.497

* Current anxiety was evaluated using four levels from 0 (null) to 3 (extreme) and is reported when present (1 or 2 or 3). ** Pain was evaluated using an 11-point numerical rating scale (NRS) from 0 = no pain to 10 = maximum imaginable pain. *** A neuropathic pain was defined by a DN4 score ≥ 4 [11]. **** Classification according the World Health Organization analgesic ladder. Number of available data in case of missing data is expressed as {xx}. Values are expressed as median (25th percentile–75th percentile) or n (%) as appropriate.

**Table 2 jcm-11-05152-t002:** Intraoperative variables in posterolateral and axillary groups.

	Posterolateral Thoracotomy n = 92	Axillary Thoracotomy n = 89	*p* Value
Duration of anesthesia, minutes	158 (136–195) {85}	166 (139–195) {88}	0.551
Duration of surgery, minutes	103 (78–129) {84}	109 (82–133) {88}	0.623
Surgical procedure			0.153
Lobectomy	57 (62.0)	64 (71.9%)	0.160
Wedge resection	23 (25.0)	19 (21.3%)	0.600
Pneumonectomy	7 (7.6)	1 (1.1%)	0.065
Other procedure	5 (5.4)	5 (5.6%)	1
Right side of surgery	46 (50.0)	52 (58.4)	0.297
Surgical retractors	92 (100.0)	59 (67.8) {87}	<0.0001
Rib fracture	22 (24.7) {89}	7 (8.2) {85}	0.004
Number of pleural drains			0.600
One	23 (25.0)	19 (21.3%)	
Two	69 (75.0)	70 (78.6%)	
Serratus preservation	83 (92.2) {90}	73 (85.9) {85}	0.226
Latissimus dorsi preservation	6 (6.8) {88}	65 (76.5) {85}	<0.0001
Transcostal suture	3 (3.4) {88}	48 (55.8) {86}	<0.0001
Extrapleural detachment	16 (18.0) {89}	7 (8.4) {83}	0.076

Number of available data in case of missing data is expressed as {xx}. Values are expressed as median (25th percentile–75th percentile) or n (%) as appropriate.

**Table 3 jcm-11-05152-t003:** Postoperative hospital stay variables in posterolateral and axillary groups.

	Posterolateral Thoracotomy n = 92	Axillary Thoracotomy n = 89	*p* Value
Type of postoperative analgesia			0.101
Thoracic epidural analgesia	82 (89.1)	77 (86.5)	0.591
Paravertebral block	7 (7.6)	3 (3.4)	0.330
No locoregional procedure	3 (3.3)	9 (10.1)	0.078
Chest tube in place at day 6 (±1)	18 (19.6)	21 (23.6)	0.589
Pain at day 6 (±1 day) *			
Pain score at rest	1 (0–2)	1 (0–2)	0.867
Pain score for cough	4 (2–6) {90}	3 (2–5) {87}	0.019
Pain score for mobilization of the ipsilateral shoulder	3 (2–4)	2 (1–4)	0.035
Neuropathic pain **	6 (6.5)	4 (4.5)	0.747
Height of hypoesthesia area around the scar, cm			
On the breast line	0 (0–8)	0 (0–5) {87}	0.722
On the axillary line	0 (0–5)	0 (0–4) {88}	0.354
On the tip of the scapula	0 (0–0) {91}	0 (0–0) {88}	0.016
Scar length, cm	17 (14–19) {91}	8 (6–11) {84}	<0.0001
Satisfaction score ***	8 (7–9) {91}	9 (8–10) {87}	0.075
Postoperative complications ≥ IIIa ****	3 (3.3)	6 (6.7)	0.325
Postoperative hospital stay, days (n)	8 (7–12)	8 (7–11)	0.132

* Pain was evaluated using an 11-point numerical rating scale (NRS) from 0 = no pain to 10 = maximum imaginable pain. ** Neuropathic pain was defined by a DN4 score ≥ 4 [11]. *** Satisfaction was evaluated using an 11-point numerical rating scale (NRS) from 0 = totally disappointed to 10 = totally satisfied. **** Classification of postoperative complications according to the Clavien classification [13]. Number of available data in case of missing data is expressed as {xx}. Values are expressed as median (25th percentile–75th percentile) or n (%) as appropriate.

**Table 4 jcm-11-05152-t004:** Patient characteristics at inclusion for each trajectory group.

	Trajectory 1 n = 54	Trajectory 2 n = 59	Trajectory 3 n = 41	Trajectory 4 n = 27	Global *p* Value	Intergroup *p* Value
Age, years	60 (51–67)	69 (58–72)	64 (55–71)	64 (56–70)	0.049	T1 vs. T2; *p* = 0.031
Sex, Male/Female	22 (40.7)/32 (59.3)	35 (59.3)/24 (40.7)	22 (53.7)/19 (46.3)	13 (48.1)/14 (51.9)	0.251	
Body mass index, kg/m^2^	23 (21–27)	24 (22–27)	25 (22–27)	26 (22–29)	0.442	
ASA					0.073	
I	8 (14.8)	4 (6.8)	3 (7.3)	3 (11.1)		
II	42 (77.8)	37 (62.7)	29 (70.7)	19 (70.4)		
III	4 (7.4)	18 (30.5)	9 (21.9)	5 (18.5)		
Cancer	42 (77.8)	47 (79.7)	33 (80.5)	23 (85.2)	0.917	
History of tobacco use	36 (73.5) {49}	38 (71.7) {53}	30 (85.7) {35}	19 (76.0) {25}	0.479	
Current anxiety *	45 (86.5) {52}	39 (68.4) {57}	28 (71.8) {39}	19 (73.1) {26}	0.131	
Painful patients	33 (61.1)	19 (32.2)	15 (36.6)	8 (29.6)	0.006	T1 vs. T2; *p* = 0.006 T1 vs. T3; *p* = 0.049 T1 vs. T4; *p* = 0.021
Pain on inclusion day **	2 (0–4) {29}	1 (0–2) {18}	2 (0–5) {15}	0 (0,1) {8}	0.306	
Mean pain during the prior week **	4 (3–6) {29}	3 (2–4) {17}	4 (2–4) {13}	4 (1–4) {8}	0.085	
Thoracic pain	7 (23.3) {30}	3 (16.7) {18}	3 (20) {15}	1 (12.5) {8}	0.947	
Neuropathic pain ***	9 (25.0) {36}	1 (2.5) {40}	2 (8.0) {25}	1 (5.6) {18}	0.015	T1 vs. T2; *p* = 0.013
Analgesic used by all patients ****	{36}	{46}	{27}	{16}		
At least one	19 (52.8)	10 (21.7)	9 (33.3)	6 (37.5%)	0.035	T1 vs. T2; *p* = 0.022
Step 1	16 (44.4)	10 (21.7)	9 (33.3)	6 (37.5%)	0.166	
Step 2	5 (13.9)	2 (4.3)	2 (7.4)	2 (12.5%)	0.409	
Step 3	1 (2.8)	1 (2.2)	0 (0.0)	0 (0.0%)	1	

* Current anxiety was evaluated using four levels from 0 (null) to 3 (extreme) and is reported when present (1 or 2 or 3). ** Pain was evaluated using an 11-point numerical rating scale (NRS) from 0 = no pain to 10 = maximum imaginable pain. *** A neuropathic pain was defined by a DN4 score ≥ 4 [11]. **** Classification according the World Health Organization analgesic ladder. Number of available data in case of missing data is expressed as {xx}. Values are expressed as median (25th percentile–75th percentile) or n (%) as appropriate.

**Table 5 jcm-11-05152-t005:** Intraoperative variables for each trajectory group.

	Trajectory 1 n = 54	Trajectory 2 n = 59	Trajectory 3 n = 41	Trajectory 4 n = 27	Global *p* Value
Axillary approach	24 (44.4)	32 (54.2)	20 (48.8)	13 (48.1)	0.781
Duration of anesthesia, minutes	162 (129–189) {52}	156 (134–198) {57}	166 (149–199) {39}	168 (135–214) {25}	0.502
Duration of surgery, minutes	102 (77–127) {52}	109 (76–130) {57}	107 (89–134) {38}	111 (79–152) {25}	0.825
Surgical procedure					
Lobectomy	35 (64.8)	39 (66.1)	29 (70.7)	18 (66.7)	0.940
Wedge resection	11 (20.4)	17 (28.8)	8 (19.5)	6 (22.2)	0.679
Pneumonectomy	3 (5.6)	1 (1.7)	3 (7.3)	1 (3.7)	0.513
Other procedure	5 (9.3)	2 (3.4)	1 (2.4)	2 (7.4)	0.444
Right side of surgery	29 (53.7)	32 (54.2)	23 (56.1)	14 (51.8)	0.993
Surgical retractors	48 (88.9)	46 (80.0)	33 (82.5) {40}	24 (92.3) {26}	0.283
Rib fracture	8 (14.8)	10 (17.9) {56}	3 (7.9) {38}	8 (30.8) {26}	0.121
Number of pleural drains					0.884
One	12 (22.2)	15 (25.4)	8 (19.5)	7 (25.9)	
Two	42 (77.8)	44 (74.6)	33 (80.5)	20 (74.1)	
Serratus preservation	47 (88.7) {53}	51 (87.9) {58}	36 (94.7) {38}	22 (84.6) {26}	0.595
Latissimus dorsi preservation	19 (36.5) {52}	28 (48.3) {58}	17 (45.9) {37}	7 (26.9) {26}	0.244
Transcostal suture	13 (25.5) {51}	21 (36.2) {58}	14 (35.9) {39}	3 (11.5) {26}	0.082
Extrapleural detachment	7 (13.5) {52}	7 (12.3) {57}	6 (16.2) {37}	3 (11.5) {26}	0.947

Number of available data in case of missing data is expressed as {xx}. Values are expressed as median (25th percentile–75th percentile) or n (%) as appropriate.

**Table 6 jcm-11-05152-t006:** Postoperative hospital stay variables for each trajectory group.

	Trajectory 1 n = 54	Trajectory 2 n = 59	Trajectory 3 n = 41	Trajectory 4 n = 27	Global *p* Value	Intergroup *p* Value
Type of postoperative analgesia					0.788	
Thoracic epidural analgesia	48 (88.9)	52 (88.1)	34 (82.9)	25 (92.6)	0.707	
Paravertebral block	4 (7.4)	2 (3.4)	3 (7.3)	1 (3.7)	0.736	
No locoregional procedure	2 (3.7)	5 (8.5)	4 (9.8)	1 (3.7)	0.615	
Chest tube in place at day 6 (±1)	13 (24.1)	13 (22.0)	7 (17.1)	6 (22.2)	0.880	
Pain at day 6 (±1 day) *						
Pain score at rest	2 (0–3)	0 (0,1)	1 (0–2)	1 (0–3)	<0.0001	T1 vs. T2; *p* < 0.0001 T2 vs. T4; *p* = 0.024
Pain score for cough	5 (4–7) {51}	2 (1–3)	4 (2–5)	5 (3–6) {26}	<0.0001	T1 vs. T2; *p* < 0.0001 T1 vs. T3; *p* = 0.005 T2 vs. T3; *p* = 0.034 T2 vs. T4; *p* < 0.0001
Pain score for mobilization of the ipsilateral shoulder	4 (2–6)	2 (1–3)	3 (2–4)	4 (2–4)	<0.0001	T1 vs. T2; *p* < 0.0001 T2 vs. T4; *p* = 0.018
Neuropathic pain **	5 (9.3)	1 (1.7)	2 (4.9)	2 (7.4)	0.296	
Height of hypoesthesia area around the scar, cm						
On the breast line	1.5 (0–7.7) {54}	0 (0–4) {58}	0 (0–7) {41}	0 (0–8.5) {26}	0.353	
On the axillary line	0 (0–3) {54}	0 (0–5) {58}	0 (0–5) {41}	3 (0-5.5) {27}	0.413	
On the line tip of the scapula	0 (0–0) {54}	0 (0–0) {57}	0 (0–0) {41}	0 (0–0) {27}	0.674	
Scar length, cm	15 (9–19) {54}	12 (7–16) {54}	12 (8–17) {41}	14 (13–18) {26}	0.047	ns
Satisfaction score ***	9 (8,9) {53}	9 (8–10) {58}	8 (7–10) {40}	9 (6–10) {27}	0.183	
Postoperative complications ≥ IIIa ****	2 (3.7)	1 (1.7)	4 (9.8)	2 (7.4)	0.247	
Postoperative hospital stay, days (n)	8 (7–13)	9 (7–11)	9 (6–13)	8 (7–11)	0.840	

* Pain was evaluated using an 11-point numerical rating scale (NRS) from 0 = no pain to 10 = maximum imaginable pain. ** Neuropathic pain was defined by a DN4 score ≥ 4 [11]. *** Satisfaction was evaluated using an 11-point numerical rating scale (NRS) from 0 = totally disappointed to 10 = totally satisfied. **** Classification of postoperative complications according to the Clavien classification [13]. Number of available data in case of missing data is expressed as {xx}. Values are expressed as median (25th percentile–75th percentile) or n (%) as appropriate. ns = not statistically significant.

**Table 7 jcm-11-05152-t007:** Multivariable analysis of factors linked to pain trajectories (trajectory group 2 as reference).

	Trajectory Group 1 OR (CI95%)		Trajectory Group 3 OR (CI95%)		Trajectory Group 4 OR (CI95%)	
Age	0.96	(0.91–1)	0.99	(0.94–1.03)	0.98	(0.93–1.04)
ASA class						
II	0.15	(0.01–2.36)	0.57	(0.03–13.01)	0.21	(0.01–6.07)
III	0.02	(0.001–0.52)	0.51	(0.02–13.78)	0.06	(0.001–2.48)
Anxiety	4.75	(0.90–25.14)	0.75	(0.21–2.59)	1.91	(0.38–9.64)
Preoperative pain	6.94	(1.54–31.27)	1.46	(0.40–5.34)	0.78	(0.13–4.56)
At least one analgesic	1.58	(0.39–6.44)	1.85	(0.46–7.48)	2.58	(0.40–16.72)
Rib fracture	0.86	(0.15–4.78)	0.13	(0.01–1.33)	2.51	(0.49–12.85)
Transcostal suture	0.59	(0.14–2.54)	1.49	(0.39–5.73)	0.32	(0.05–2.08)
Scar length	1.20	(1.05–1.38)	1.11	(0.98–1.27)	1.12	(0.96–1.31)

OR (CI95%): odds ratios and 95% confidence intervals.

## Data Availability

Data supporting the reported results are available on the Dryad website open-access repository (https://doi:10.5061/dryad.f4qrfj6zp, accessed on 26 August 2022).

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
