# Peer review of "Early Postoperative Pain Trajectories after Posterolateral and Axillary Approaches to Thoracic Surgery: A Prospective Monocentric Observational Study"

_jcm, 2022, doi:10.3390/jcm11175152_

Round 1
Reviewer 1 Report
Dear Authors:
Thank you for submiting this interesting paper.
Pain treatment is a paramount aspect in thoracic surgery. Indeed, last year's most improvements in thoracic surgery have been directed to reduce the inherent aggression of this kind of treatments, looking for a less painful surgery.
Your work is a very well structured paper, but it completely lacks of interest nowadays. There are plenty of old studies comparing types of thoracotomies with conclusions not much different tan yours. Besides, it takes years since most of thoracic surgery departments around the world try to perform the most of their surgeries through VATS approach, so nowadays, it's not worthy another study on pain control in thoracotomies, since on the one hand, they have previously been done in great number, and on the other hand, thoracotomies are no more the desired approach for thoracic pathology.
It happens the same with the pain management strategy; in the same line of trying to reduce aggression, epidural remains as an out-dated pain control technique, in favor of less aggressive techniques like paravertebral catheters.
Besides, the whole work is based in a bias; the chosing criteria of the type of incision is based on surgeon's experience, so this will always change not just the approach, but the whole surgery, not being thus comparable one technique with the other.
Author Response
We agree with many of these comments and have shared them in the article:
- The reviewer wrote: “most thoracic surgery departments around the world try to perform the most of their surgeries through VATS approach”. We wrote in the Introduction: “Surgical techniques have been oriented towards less damaging incisions (muscle-sparing posterolateral thoracotomy, axillary or anterior approaches) or mini-invasive procedures (video-assisted thoracic surgery (VATS) and robot-assisted thoracic surgery (RATS) [3,4].” The chapter “4.4. Generalizability” shows that we have warned potential readers of this fact: “Our study combined patients having had a large surgical approach and a mini-invasive one. These two approaches being progressively discarded and replaced by video-assisted thoracoscopic surgery and robot-assisted thoracic surgery; our results cannot be generalized to the whole lung surgical population.” In addition, we underline that some clinical situations are still not adapted to mini-invasive approach and require a traditional thoracotomy and that the axillary thoracotomy technique is close to VATS.
- The reviewer wrote: “It happens the same with the pain management strategy; in the same line of trying to reduce aggression, epidural remains as an out-dated pain control technique, in favor of less aggressive techniques like paravertebral catheters”. We wrote in the Introduction: “Postoperative analgesia techniques have also progressed with new techniques of loco-regional analgesia which can substitute epidural analgesia (paravertebral nerve block, erector spinae plane block of the spine, …) [5].” Nevertheless, there are still indications for epidural analgesia. This was the choice of our team at the time of this study. We have modified the chapter “4.4. Generalizability” to insist on the fact that thoracic epidural analgesia is no longer the rule given the evolution of surgical technique. The modified text is: “Moreover, most of our patients benefited from a thoracic epidural postoperative analgesia. This was the choice of our team at the time of this study but this strategy has evolved towards less invasive technique [5, 6].”
Conversely, we wish to defend our work:
- This is the first study concerning the analysis of postoperative pain trajectories and this study highlights the importance of some predictive factors of severe and permanent postoperative pain (existence of preoperative pain and scar length) and a protective factor (ASA class III). Interestingly, preoperative pain has an OR which is three times greater than the scar length. This suggests that we should be more aggressive in treating preoperative pain.
- The choice of surgical technique by the surgeon obviously induces a bias but most studies comparing posterolateral thoracotomy and VATS do not include randomization. Please note that our revised text mentioned: “Our study did not include randomization for the choice of the approach, because the surgeons considered it was unethical to use approaches they were unfamiliar with. It was decided that participating surgeons would use the approach they routinely used and were familiar with. Such conduct is usual since most studies comparing PL thoracotomy and VATS do not include randomization.”
Reviewer 2 Report
Thank you very much for selecting me as a reviewer for the article, entitled “EARLY POSTOPERATIVE PAIN TRAJECTORIES AFTER POSTEROLATERAL AND AXILLARY APPROACHES FOR 1 THORACIC SURGERY: PART 1 OF A PROSPECTIVE MONOCENTRIC OBSERVATIONAL STUDY”
In accordance with the guidelines for reviewers, I would like to describe some comments shown below for authors of this research.
1. What is the main question addressed by the research?
The background of this research is that effective postoperative analgesia plays a major role in the prevention of major morbidity and mortality after thoracotomy, and that a better understanding of postoperative pain could make it possible to treat or even prevent it and therefore improve the postoperative experience.
The main question addressed by the research is to investigate the pain trajectory, which makes it possible to classify a patient into a specific cluster and to identify potential factors related to the worst" trajectory.
2. Do you consider the topic original or relevant in the field, and if so, why?
I consider that the topic of the research is original because the concept of pain trajectory is unique and uncommon among thoracic surgeons.
As the authors suggest in the study that the present research is the first description of pain trajectories during the early postoperative period after lung surgery.
3. What does it add to the subject area compared with other published material?
What the research adds to the subject is the understanding acute postoperative pain as a trajectory rather than as one or more simple point estimates of intensity.
4. What specific improvements could the authors consider regarding the methodology?
I consider the methodology of the research to be well summarized and written.
5. Are the conclusions consistent with the evidence and arguments presented and do they address the main question posed?
The authors have come to the conclusion that four trajectories for early postoperative pain after lung surgery have been identified and that the major risk factor for permanent pain trajectory is preoperative pain. They have successfully addressed the main question posed in the study.
6. Are the references appropriate?
I consider that the references cited are appropriate.
7. Please include any additional comments on the tables and figures.
In my viewpoint, the tables and figures in the manuscript are well summarized.
Author Response
We thank the reviewer for his comment.
Reviewer 3 Report
This is a interesting retrospective study regarding the postoperative pain after different kind of thoracotomy. I think that this is a well conducted study, the methods and the results are well reported. The introduction section can be improved with a reference to the other analgesic methods after thoracic surgery (es. crioanalgesia).
Author Response
We thank the reviewer for his comment. As suggested, we have added cryoanalgesia as an alternative analgesic technique and a reference (Humble et al.) which completes Gupta et al.